# Empirical Model of Gaseous Mercury Emissions for the Analysis of Working Conditions in Outdoor Highly Contaminated Sites

Rafael Rodríguez [1,*], Hector Garcia-Gonzalez [2] and Efrén García-Ordiales [1]

1   Department of Mining Exploitation and Prospecting, School of Mining, Energy and Materials Engineering, University of Oviedo, Independencia 13, 33004 Oviedo, Spain
2   Instituto Nacional de Silicosis, 33011 Oviedo, Spain
*   Correspondence: rrodrifer@uniovi.es; Tel.: +34-985104253

**Highlights:**

**What are the main findings?**

- Gaseous mercury concentration was measured in a strongly contaminated mercury mining and metallurgy site.
- An empirical model was developed to predict the gaseous mercury concentration at any temperature.

**What is the implication of the main finding?**

- Occupational risk of inhalation of mercury according to the standard EN-689:2018 can be assessed with the model.
- Works in the site can be carried out without restriction for temperatures under 15 °C.

**Abstract:** The project SUBproducts4LIFE is a LIFE-funded research project that aims to demonstrate innovative circular economy concepts by repurposing industrial subproducts/waste (coal ash and gypsum from coal power plants, blast furnace slag, and steelmaking slag from steel factories) for the remediation of contaminated soils and brownfield areas associated with Hg mining on a large scale. Within the project, one of the objectives is related to worker safety by developing protocols and elaborating a guide of good practices to work under in these highly contaminated areas. The present research aims to assess the working conditions in an abandoned mine facility in terms of gaseous mercury in the environment, evaluating the main areas of the mine to ensure the health and safety of all workers and visitors. The study developed an empirical model for estimating the gaseous mercury concentration at any temperature with the target of scheduling the work to avoid occupational hazards. The research concluded that working without restrictions for temperatures under 15 °C in all the mine's facilities is possible.

**Keywords:** mercury; abandoned mines; airborne; spoil tip; risk management

## 1. Introduction

Spain has the most extensive mercury production in all history; it is estimated that approximately one-third of the world's mercury production was mined in Spain [1], and the main mine was Almaden (Ciudad Real). However, there were other mercury mines in other regions, such as Castilla Leon and Asturias. Most of these mines were abandoned without any restoration plan; nowadays, most are an important environmental problem. Researchers are looking for innovative solutions to deal with these sites. Dealing with mercury-contaminated facilities with high mercury emissions could be hazardous to work-ers' health. This paper assesses the working conditions in an abandoned mercury mine facility regarding mercury emissions hazards.

### 1.1. Mercury Contamination in Abandoned Mining Facilities

The high level of contamination by heavy metals in abandoned mines is a severe environmental problem for public health. Precisely because of the problem's seriousness, a vast scientific bibliography has investigated this topic.

An objective of this research has been the protection of the environment. In recent decades, considerable progress has been made in understanding mercury gas emissions from Hg-enriched areas (e.g., Gustin [2]; Feng et al. [3]; and Wang et al. [4]). The results have shown that the mercury gas emission rates from Hg-enriched areas are much greater than the values measured in the background area, and the contribution of gaseous mercury to the atmosphere from Hg-enriched soil in the mercuriferous belt was vastly underestimated [2,3]. In this way, there are topics that have been extensively studied, e.g., Hg concentration in the soil and water surrounding contaminated sites ([5–7]) and emission rates into the atmosphere and the distribution of contamination surrounding sites contaminated by mercury mining [4,8–13]. Another research line is the development of emission models which help to analyze the potential risks related to Hg contamination, for example, the studies by Lindberg et al. [14] and Llanos et al. [15]. Even other very particular studies, such as the transfer of Hg to plants, have been studied by Matanzas et al. [16].

Another research line is the influence of contamination on the health of people, such as both of the general studies Kim et al. [17] and Wu et al. [18] and the more specific studies on human diseases undertaken by Phelps et al. [19] and Koenigsmark et al. [20].

Lastly, there is another type of research related to the remediation of contaminated sites, and it is more concretely associated with the occupational risks of working in these areas. Nevertheless, investigations in this field are scarce. However, there have been very relevant studies carried out recently, for example, those by Eckley et al. [21], Wcislo et al. [22,23], and Wu et al. [18].

### 1.2. Effects of Mercury on Human Health and Regulations for Worker Exposure to Mercury

According to the WHO (World Health Organization), mercury inhalation can harm the nervous, immune, and digestive systems, as well as the lungs and kidneys, and can be fatal. Some side effects of exposure include memory loss, neuromuscular effects, headaches, cognitive and motor dysfunction, tremors, and insomnia. Some forms of mercury have been shown to cause a variety of tumors in rats and mice at extremely high concentrations [24–26].

Methylmercury and metallic mercury vapor are the most toxic forms of mercury, and exposure to high levels can permanently harm the brain and kidneys and the developing fetus. Abdominal pain, inflammatory bowel disease, ulcers, bloody diarrhea, destruction of intestinal flora, endocrine system affections, and reduced fertility are mercury's other effects. Mercury is bioaccumulative in the body, with the primary sites being the liver, brain, and kidneys [27]. The human body retains approximately 80% of any inhaled mercury, and it accumulates in the brain and other internal organs [28].

The Minamata Convention on Mercury (http://www.mercuryconvention.org/ accessed on 26 April 2022) is now an international program prohibiting the trade and use of mercury.

The Spanish Instituto Nacional de Seguridad e Higiene en el Trabajo (INSST) provides legal health and safety standards for workers in Spain. The occupational exposure limit value (OELV), which is the average exposure for each chemical, is based on an eight-hours-per-day, 40-h-per-week work schedule, and for mercury, the OELV is 0.02 mg/m$^3$ (20,000 ng/m$^3$).

Short-term exposures may be increased up to three times the OELV for 15 min, as a maximum, on no more than four occasions in an eight-hour working day, with a minimum gap of one hour between two consecutive peak exposures. It should never be more than five times the OELV value.

Furthermore, the OELV 8 h limit should not be exceeded during working hours.

Examples of OELVs are the threshold limit value time-weighted average (TLV-TWA) identified by the American Conference of Governmental Industrial Hygienists (ACGIH) and the limites de exposición profesional (LEP) identified by the INSST in Spain.

### 1.3. Objectives in the Framework of SUBproducts4LIFE

SUBproducts4LIFE is a research project co-funded by the European Union as part of the LIFE program. The project SUBproducts4LIFE aims to demonstrate innovative circular economy concepts by repurposing industrial waste (coal ash and gypsum from coal power plants, blast furnace slag, and steelmaking slag from steel factories) for the remediation of contaminated soils and brownfield areas associated with mercury mining on a large scale [24].

This work was a follow-up to a previous one on the characterization of arsenic and mercury contamination in the air. The current investigation focused primarily on gaseous mercury because the particle concentrations of As and Hg in the air were low in the previous study [24]. As a result, the following points can be shown:

- The study's primary goal was to assess the working conditions in an abandoned mine facility regarding gaseous mercury in the environment.
- The approach utilized to characterize the site is given, and some data can be used to build work protocols in the most troublesome areas.
- It was a research project focused on the prevention of occupational hazards.
- It was a macroscopic study that does not detail the physical and chemical processes.
- The monitoring was centered on determining the mercury concentration in the air, the most critical factor in occupational health and safety.
- The research was conducted in a constrained environment (the workplace), with a maximum measurement distance of 150 m.

Therefore, the scale is intermediate; this was not a detailed examination of what was going on beneath the rubble (i.e., it was not a laboratory test or a study of cells), nor was this environmental research (where the distances would be kilometers).

No extensive investigation of the debris was completed, and all calculations were based on the mercury in the atmosphere. According to Agnan [29], and as referenced by Horvart and Kotnit [30], there is a correlation between the concentration of mercury in the soil and emissions; however, as Johnson et al. showed, the flow of mercury to the atmosphere cannot be determined only from the mercury concentration in the ground [31]. Schlüter suggested that Hg moves over short distances by a diffusion mechanism [32]. Because the range of variables is limited, it is possible to design a non-complex empirical model that can be used in conjunction with more complicated models that account for all of the process variables, such as those proposed by Zhang and Lindberg or Sholtz [8,33].

A simple empirical model is defined in the following, and it is a process for taking a set of data that allows the model's parameters to be estimated. In this case, a strong correlation between emissions and ambient temperature was found, similar to that found by Scholtz [8] (referencing Lindberg et al. [14], Siegel and Siegel [34], and Zhang et al. [35]).

The study is significant because it confirms that models generated through laboratory experiments or gas diffusion cells are applicable through a macroscopic investigation. In this situation, the concentration is measured not as an experiment but as a preventive measure related to occupational risks. On the other hand, it is quite valuable for preliminary job planning.

The high level of contamination by heavy metals in abandoned mines is a severe environmental and public health problem. Administrations, companies, and institutions must provide the necessary means to remedy these lands. Improving the conditions of these highly degraded sites requires carrying out work that exposes workers to this contamination.

Before carrying out these works, a preliminary investigation is necessary to establish the type of contamination, the contaminating agents, the different degrees of contamination

in the other areas of the mining facility, and the risks associated with working in these contaminated areas.

This article describes how this initial study regarding gaseous mercury was carried out in the case of La Soterraña. These actions are proposed as a protocol or guide to follow in other similar initial research works in other contaminated places. There were three main objectives:

- The preliminary research study has been analyzed as one more task, and it must comply with current legislation.
- Some criteria have been established for taking samples. An essential database has been obtained for the concentration levels of gaseous mercury and its distribution in the site with different ambient temperatures.
- An empirical model has been developed that makes it possible to predict the concentration of gaseous mercury as a function of temperature, which can be used for planning work and analyzing the potential risk of exposure to workers.

## 2. Material and Methods

### 2.1. La Soterraña Mine

The Soterraña mine is located 30 km south of Oviedo, 5 km northwest Pola de Lena, in Asturias, Spain. The mine was in operation between 1948 and 1974. It was one of the most critical mercury mines in the north of Spain.

The mine's geology is a low-temperature hydrothermal epigenetic deposit. The predominant minerals are cinnabar (mercury sulfide), realgar (arsenic sulfide), and, in smaller proportion, orpiment (another arsenic sulfide). Arsenopyrite, marcasite, and pyrite are also hosted in fractured limestones and shales [36]. The gangue is composed of carbonates, quartz, and argillaceous minerals (kaolinite and dickite) [10].

### 2.2. Lumex RA-915

The instrument sampling the airborne mercury was a Lumex RA-915 (Lumex instruments, Fraserview Place, Mission, BC, Canada) with a 1–100,000 ng/m$^3$ analytical gaseous mercury range. The apparatus takes 10 L of air each minute and reports one analysis every second, storing the data in the internal data logger. It was designed to work between 1 °C and 40 °C [37]. The Lumex RA-915 fulfils all the requirements outlined in the standard EN 482:2021 "Workplace exposure—Procedures for the determination of the concentration of chemical agents—Basic performance requirements" [38].

This device has been widely used to monitor gaseous mercury in various situations and environments in the scientific literature and by reference organizations. During the first campaigns, a GPS Garmin Etrex Touch 35 (Garmin International Inc, Olathe, KS, USA) was utilized with the mercury analyzer. The GPS was designed to record positions every second to correlate this data with the Hg measurements.

The LUMEX RA-915 analyses gaseous mercury, omitting mercury particles eliminated via a filter at the equipment's inlet. On every sampling day, the device was warmed up for at least 20 min before the first reading, following the manufacturer's instructions.

### 2.3. Sampling Procedure

The design of a measurement campaign requires defining several aspects, such as selecting the points to measure gaseous mercury (number of control points, location, and height), the duration of the measurements, and how often a measuring campaign has to be carried out.

There are no fixed values of these parameters that serve all situations. Those used here were the result of experience and the specifics of this case. In all the campaigns, the measurements started from the outside of the mine facilities and moved to the more contaminated areas. Regarding the number of control points, it was established that there should be at least one point where the work was to be carried out in each area. Another criterion was to increase the distance between points in the places with low contamination

and to reduce it around possible foci to identify them. Lastly, the number of points should have allowed for running a campaign in approximately 1 h. Thus, the 22 control points and their location were defined.

The measurement time at a single point was set to 2 to 5 min. First, because they were based on our experience, measurements of 2 to 5 min were sufficiently representative (this was checked with measurements of a 2 h duration at each point), similar to the trial periods used in other cases described in the literature [39,40]. In a previous study of the area, it was better to make a representative measurement of all the points in a short space so that the weather conditions (or other factors) varied as little as possible. If we had set this duration to be approximately one hour, we should not have spent more than 2–5 min in each of the 22 points.

Regarding the measurement height, OELVs were considered for measurements in the worker "breathing zone". Nevertheless, the measurements were carried out at 1.0–1.5 m above ground level because of the recommended height for airborne environmental values [41]. Due to airborne mercury concentrations tending to diminish with altitude, taking measurements at a lower height was safe with respect to health and safety.

Several authors have demonstrated the relationship between the flow of mercury from a solid surface and the temperature, including Lindberg et al. [14], Siegel and Siegel [34], Zhang et al. [35], and Scholtz et al. [8]. For this reason, the measurement campaigns were carried out at different seasons of the year to obtain data for varying temperatures in the range between 5 °C and 30 °C, which is the range of typical temperatures in the region (exceptionally, there may be days with temperatures of below 5 °C and higher than 30 °C, but this would only be a few days per year).

Airborne mercury levels in the area were recorded in previous studies [10]. For health and safety concerns, gaseous mercury readings were collected from time to time to guarantee that work could be completed without high mercury levels on the site.

However, in this research, a sample procedure was developed based on the assumption that no prior information about the location existed. Furthermore, the monitoring campaigns were designed systematically to obtain the most accurate information possible to define the site and design work protocols. As a result, a route was established that included 22 control points throughout the area where measurements of gaseous mercury concentrations would be performed under various conditions (Figure 1). There were three distinct tiers: points 1 to 12 represent level 0, level 1 is represented by points 13 to 17, and level 2 is represented by points 18 to 22. The height difference between the levels is approximately 10 m. Levels 0 and 2 are where the SUBproducts4LIFE project's work took place.

Although the study's findings can be interpreted in terms of environmental contamination, the study's primary goal was to characterize the area to carry out restoration and remediation work or prevent occupational hazards.

As a result, a study of how to safely conduct monitoring and control duties was undertaken.

The first recommendation is to begin the survey far away from the most contaminated places to guarantee that gaseous mercury concentrations are low, and then proceed toward the most-polluted and highest-risk zones. On the other hand, a measurement was made at each place for a time ranging from 2 to 5 min, depending on the observed fluctuations. In the same way, the route would not run for more than one hour to reduce exposure.

It is important to note that Spanish law restricts permanence in highly hazardous areas. In an eight-hour working day, a worker could stay in one place with three times the OELV = 60,000 ng/m$^3$ for a maximum of 15 min on no more than four occasions. It is possible to reach very high average airborne mercury readings in areas containing demolition debris from the metallurgical plant, exceeding the 20,000 ng/m$^3$ limit. It was established that the monitoring time in these areas would be decreased to 2 min to avoid the risk of mercury inhalation.

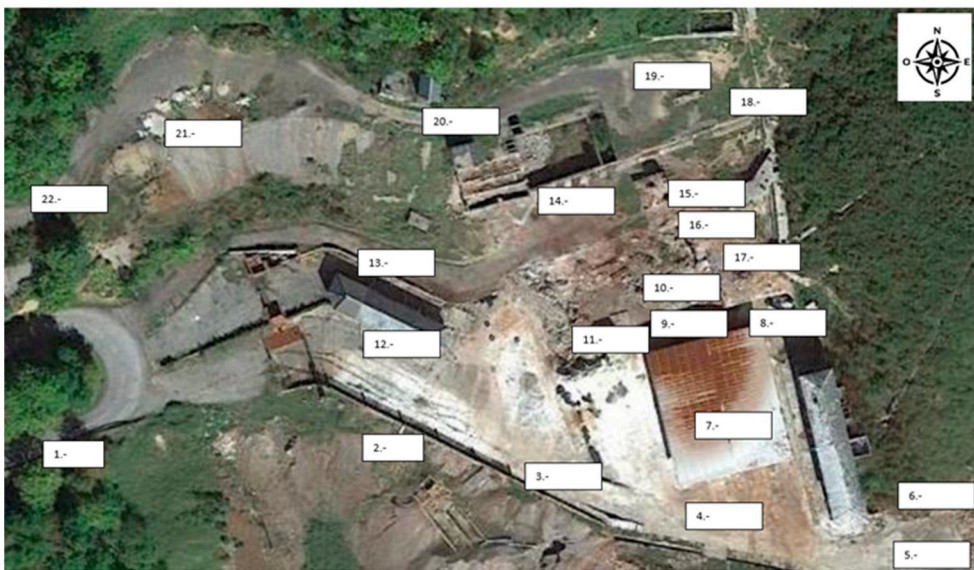

**Figure 1.** Sampling route design (points 1 to 12 are at level 0, points 13 to 17 are at level 1 and points 18 to 22 are at level 2).

The first measurements were obtained at approximately 1.5 m, while the following measurements were taken at a lower height of 1.0 m. However, it was determined that the height difference had no significant impact on the measures.

Airborne mercury concentrations tend to diminish with height; thus, taking measurements at a lower height was safe with respect to health and safety. Figure 2 shows the Lumex analyzer at points 10 and 7.

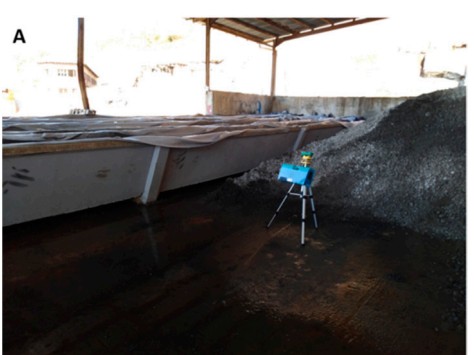
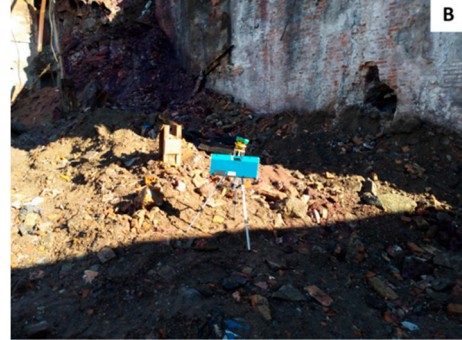

**Figure 2.** Sampling points 7 (**A**) and 10 (**B**).

In order to establish a range of emission values for the site, it was decided to measure gaseous mercury concentrations from lower to higher temperatures because gaseous mercury emissions are temperature dependent. When no data is available, starting monitoring campaigns on cold days is best to allow for starting with the safest settings. Because historical information was available at moderate and low temperatures, the monitoring campaigns began in the most extreme conditions.

The first two surveys were made in August when temperatures ranged from 30 to 29 degrees Celsius, which is extremely hot for Asturias. Shortly after, the operation was repeated in October on a day deemed cold, with a temperature of 6.5 °C below the Asturias average. As a result, the limit values of the gaseous mercury concentration variation range as a function of temperature could be determined at all places along the route.

The campaigns had to be run at different seasons of the year to obtain measurements at varying temperatures. As a result, the campaigns were carried out on warm days, when

no work was completed in the mining facility, and on cooler days, when some work was completed, as described below.

The gaseous mercury averages acquired while workers were on-site could be used to monitor working conditions.

## 3. Results and Discussion

### 3.1. Gaseous Mercury Distribution on the Site

3.1.1. Overall Results

The 15 gaseous mercury concentration surveys are presented in four tables based on the four distinct seasons when the measurement campaigns were conducted.

Table 1 displays the findings from the first four surveys published by Garcia-Gonzalez et al. [24]. They were completed before the start of the SUBproducts4LIFE project's work on contaminated solids and liquids treatment.

**Table 1.** The first campaign of surveys.

| | Campaign | Survey 1 | Survey 2 | Survey 3 | Survey 4 |
|---|---|---|---|---|---|
| | Temperature (°C) | 29 | 30 | 15 | 6.5 |
| | Location | Concentration Gaseous Mercury in the Air (ng/m$^3$) | | | |
| 1 | Road | 32 | 10 | 200 | 67 |
| 2 | External wall | 385 | 250 | 425 | 2379 |
| 3 | Store | 339 | 1750 | 950 | 927 |
| 4 | Trench | 742 | 450 | 350 | 762 |
| 5 | Yard extension | 360 | 75 | 1900 | 601 |
| 6 | Yard | 53 | 275 | 25 | 11 |
| 7 | Shed | 296 | 800 | 3000 | 223 |
| 8 | Shed corner | 2317 | 1500 | 2000 | 2830 |
| 9 | Bottom rubble furnace | 20,867 | 15,000 | 30,000 | 5524 |
| 10 | Rubble (metallurgy furnace debris) | 58,488 | 50,000 | 50,000 | 15,827 |
| 11 | Internal wall | 3305 | 17,500 | 17,500 | 6856 |
| 12 | Office | 437 | 250 | 800 | 1437 |
| 13 | First level | 231 | 700 | 175 | 873 |
| 14 | First level wall | ND | 800 | 4000 | 3238 |
| 15 | First level rubble 1 | 3855 | 4500 | 2500 | 4936 |
| 16 | First level rubble 2 | ND | 4500 | 4000 | 4097 |
| 17 | First level chimney | 768 | 6000 | 30,000 | 10,261 |
| 18 | Second level stairs | 194 | 650 | 900 | 702 |
| 19 | Second level ditch | 156 | 500 | 1100 | 564 |
| 20 | Second level transformer | 253 | 1000 | 1200 | 1005 |
| 21 | Furnace slag heap | 455 | 350 | 1400 | 263 |
| 22 | Second level entrance | 56 | 30 | 300 | 31 |

ND = not determined.

In order to collect data under the most adverse conditions, we took measurements in August during high temperatures of 29 °C and 30 °C (infrequent in Asturias), followed by two additional campaigns in September, with a temperature of 15 °C, and October, with a low temperature of 6.5 °C.

The results of the third survey, which took place at a temperature of 15 degrees Celsius, were completely unexpected. It was demonstrated in several subsequent surveys that the concentrations of gaseous mercury at that temperature were lower than those obtained that day. This fact has not been satisfactorily explained; it is possible that the reading was false due to the effects of water condensation (Asturias has a very humid climate). Another explanation is that the soil temperature was much higher than the air temperature in this case. The measure was obtained in the morning, but these days were scorching, and soil can be much hotter than air in the morning.

However, to be on the safe side, a temperature of 15 °C was set as a limit, and it was decided that work could be completed in areas with the highest concentration of gaseous mercury at temperatures above 15 °C only after a more detailed study and with assumed restrictions.

New surveys were carried out to determine the gaseous mercury concentration in the air at the low range of temperatures because work with demolition rubble should be completed at low temperatures. The data of the following three surveys are summarized in Table 2. Survey number 5 took place in November on a cold day for Asturias, with a maximum temperature of 7.5 °C.

**Table 2.** Second campaign of surveys.

|  | Campaign | Survey 5 | Survey 6 | Survey 7 |
|---|---|---|---|---|
|  | **Temperature (°C)** | **7.5** | **12.5** | **10.5** |
|  | **Location** | **Concentration Gaseous Mercury in the Air (ng/m³)** | | |
| 1 | Road | 21 | 31 | 61 |
| 2 | External wall | 34 | 87 | 156 |
| 3 | Store | 356 | 623 | 323 |
| 4 | Trench | 122 | 394 | 279 |
| 5 | Yard extension | 43 | 84 | 93 |
| 6 | Yard | ND | 26 | 48 |
| 7 | Shed | 310 | 564 | 765 |
| 8 | Shed corner | 258 | 1408 | 2307 |
| 9 | Bottom rubble furnace | 1330 | 6488 | 4689 |
| 10 | Rubble (metallurgy furnace debris) | 12,028 | 16,024 | 14,757 |
| 11 | Internal wall | 6967 | 10,357 | 8431 |
| 12 | Office | 270 | 268 | 397 |
| 13 | First level | 212 | 521 | ND |
| 14 | First level wall | 14,443 | 2003 | ND |
| 15 | First level rubble 1 | 6268 | 7001 | ND |
| 16 | First level rubble 2 | 8197 | 6526 | ND |
| 17 | First level chimney | 21,493 | 13,877 | ND |
| 18 | Second level stairs | ND | 2678 | ND |
| 19 | Second level ditch | ND | 784 | ND |
| 20 | Second level transformer | ND | 288 | ND |
| 21 | Furnace slag heap | ND | 199 | ND |
| 22 | Second level entrance | ND | 43 | ND |

ND = not determined.

The other two surveys, which took place in November and December, gathered more information on gaseous mercury conditions at the intermediate temperatures of 12.5 °C and 10.5 °C. No work was carried out on those days.

It is worth noting that a study like this would have to last several months to collect data on gaseous mercury concentrations across the full range of possible working temperatures. As a result, new data were collected in the following month's Table 3 (January and February). More data were obtained with a low temperature of 4 °C and intermediate temperatures of 14 °C and 14.5 °C.

**Table 3.** Surveys in January and February.

| | Campaign | Survey 8 | Survey 9 | Survey 10 | Survey 11 |
|---|---|---|---|---|---|
| | Temperature (°C) | 4 | 11 | 14 | 14.5 |
| | Location | Concentration of Gaseous Mercury in the Air (ng/m³) | | | |
| 1 | Road | 20 | 87 | 80 | 31 |
| 2 | External wall | 92 | 673 | 77 | 1096 |
| 3 | Store | 510 | 350 | 74 | 2164 |
| 4 | Trench | 415 | 151 | 315 | 7893 |
| 5 | Yard extension | 62 | ND | 178 | 299 |
| 6 | Yard | ND | 271 | 103 | ND |
| 7 | Shed | 200 | 2268 | 604 | 3215 |
| 8 | Shed corner | 1177 | ND | 5879 | 4868 |
| 9 | Bottom rubble furnace | 7205 | 7153 | 6426 | 11,493 |
| 10 | Rubble (metallurgy furnace debris) | 6512 | 15,945 | 12,089 | 25,500 |
| 11 | Internal wall | 7204 | 9107 | 8294 | 8852 |
| 12 | Office | 58 | 81 | 120 | 1160 |
| 13 | First level | 243 | 103 | 365 | 348 |
| 14 | First level wall | 2755 | 3003 | 7285 | 14,740 |
| 15 | First level rubble 1 | 4038 | 8659 | 7568 | 14,130 |
| 16 | First level rubble 2 | 8530 | 2840 | 5835 | 10,724 |
| 17 | First level chimney | 10,424 | 5035 | 17,325 | 27,232 |
| 18 | Second level stairs | 2533 | 1420 | 1524 | 7426 |
| 19 | Second level ditch | 263 | 694 | 1361 | 1096 |
| 20 | Second level transformer | 138 | 126 | 237 | 595 |
| 21 | Furnace slag heap | 222 | 164 | 400 | 789 |
| 22 | Second level entrance | 60 | 35 | 442 | 96 |

ND = not determined.

The latter allowed researchers to show that the results of control survey 3, which was conducted at 15 degrees Celsius, were, indeed, abnormal.

Another four surveys were conducted during February and March (Table 4), when there were temperature variations, to round out the data set. As a result, there are data for a low temperature of 6 °C, a medium temperature of 13.5 °C, and maximum temperatures of 21 °C and 24 °C.

**Table 4.** Fourth campaign of surveys.

| | Campaign | Survey 12 | Survey 13 | Survey 14 | Survey 15 |
|---|---|---|---|---|---|
| | Temperature (°C) | 24 | 21 | 13.5 | 6 |
| | Location | Concentration of Gaseous Mercury in the Air (ng/m$^3$) | | | |
| 1 | Road | 87 | 150 | 55 | 31 |
| 2 | External wall | 172 | 2375 | 194 | ND |
| 3 | Store | 641 | 1288 | 429 | ND |
| 4 | Trench | 1133 | 805 | 386 | 402 |
| 5 | Yard extension | 356 | 472 | ND | ND |
| 6 | Yard | 39 | 51 | ND | 48 |
| 7 | Shed | 3393 | 1794 | 916 | 449 |
| 8 | Shed corner | 4977 | 6738 | ND | 664 |
| 9 | Bottom rubble furnace | 19,681 | 8785 | ND | 4538 |
| 10 | Rubble (metallurgy furnace debris) | 48,397 | 29,518 | 8890 | 11,011 |
| 11 | Internal wall | 6907 | 10,063 | 4387 | 1306 |
| 12 | Office | 976 | 3544 | ND | ND |
| 13 | First level | 229 | 1409 | ND | ND |
| 14 | First level wall | 7903 | 1395 | ND | ND |
| 15 | First level rubble 1 | 8443 | 5085 | ND | ND |
| 16 | First level rubble 2 | 13,948 | 20,005 | ND | ND |
| 17 | First level chimney | 19,775 | 6780 | ND | ND |
| 18 | Second level stairs | 2941 | 1543 | ND | ND |
| 19 | Second level ditch | 491 | 345 | 567 | ND |
| 20 | Second level transformer | 1121 | 241 | ND | 688 |
| 21 | Furnace slag heap | 345 | 348 | 1282 | 330 |
| 22 | Second level entrance | 77 | 232 | ND | ND |

ND = not determined.

Since there was evidence of points where gaseous mercury concentration was not a problem for work in previous campaigns, it was only measured at the most critical points for work with high gaseous mercury concentrations.

3.1.2. Survey Results

Figure 3A depicts an example of a continuous record of gaseous mercury along the route taken on survey 9 (January). The temperature was 11 degrees Celsius, a typical winter temperature in Asturias. The concentration was generally low (less than 2000 ng/m$^3$, 10% of the OELV) and compatible with routine work at all points along the route (taking the necessary measures).

High concentrations (approximately OELV = 20,000 ng/m$^3$ or even higher) were only found in areas with contaminated debris from the demolished metallurgical plant buildings.

The measurements in the two areas with the most contamination, the debris at level 0 (the first set of peaks), and the debris at level 1 (the second set of peaks) were perfectly distinguished in the record.

Although values close to the OELV (up to 18,000 ng/m$^3$) were reached on rare occasions, or even higher in extreme temperatures, the average concentration was much lower and more consistent with the sampling work.

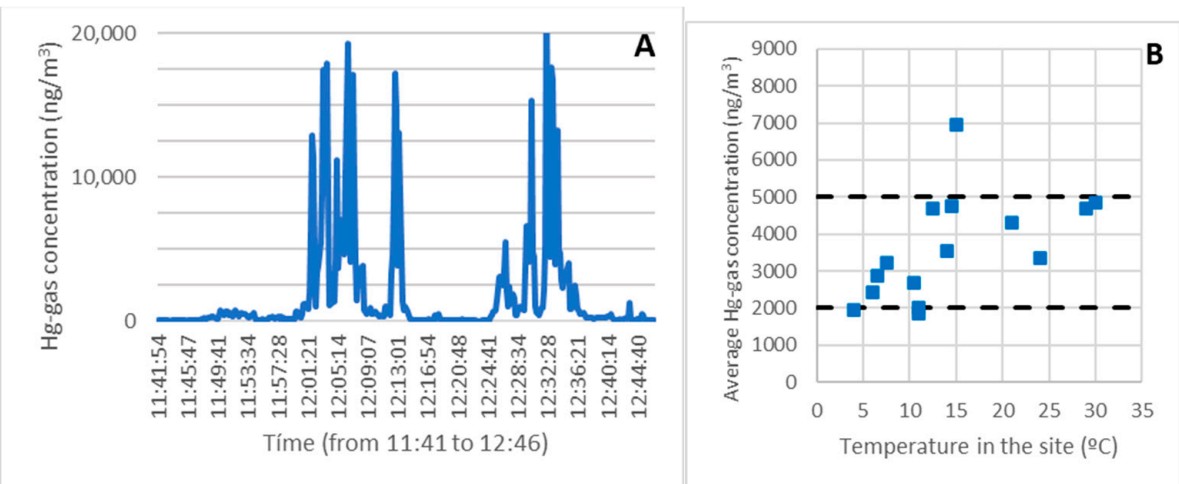

**Figure 3.** A continuous record of the gaseous mercury along the route taken on survey 9 (**A**). Average exposure as a function of temperature for the entire route (**B**).

We noted that taking measurements itself was work, and it had to be completed safely.

The average concentration obtained from the entire record was used to determine a global view of the exposure throughout the route. Similarly, the highest concentration value was taken over 15 min to obtain a value representing the highest level of exposure.

In the example route, the average exposure was 1946 ng/m$^3$, and the highest exposure was 4263 ng/m$^3$ over 15 min. Both values could be assumed, taking into account some precautions such as the use of personal protective equipment (PPE) and time limitations in various areas or points.

Figure 3B depicts the average exposure as a function of temperature for the entire route, which took approximately one hour. As can be seen, with the measures in place, the concentration was always below 5000 ng/m$^3$ throughout the route (25 percent of the OELV).

The sampling time for the survey in the most contaminated areas was reduced at high temperatures, lowering the average.

Only one data point showed a concentration level greater than 5000 ng/m$^3$, corresponding to survey three at 15 °C. Given that several other campaigns with similar or even higher temperatures had been conducted, the results of that campaign were confirmed to be anomalous and could not be considered representative. This will be discussed in more detail in the following sections.

The highest value for a single period of 15 min in one hour was 11,766 ng/m$^3$, which is five times less than the maximum allowed in exceptional cases for 15 min in one hour.

Even if that value was the weighted average for the entire hour, it would still be well below the legal limit as it would account for approximately half of the OELV. As a result, it can be concluded that the sampling was completed safely. This is significant because, as previously stated, monitoring gaseous mercury in the air is the first task to be completed in areas near contaminated rubble before any other work takes place.

### 3.1.3. Concentration Distribution on the Site

The site is depicted with an X, Y coordinate system, with the rubble of level 0 (point 10) having the coordinates of X = 150 m and Y = 70 m (Figure 4).

An indication of how the gaseous mercury was spread over the area is shown by representing the values of the gaseous mercury concentrations in the air in a coordinate system.

The distribution corresponding to survey 2 with a temperature of 30 °C is depicted in Figure 5.

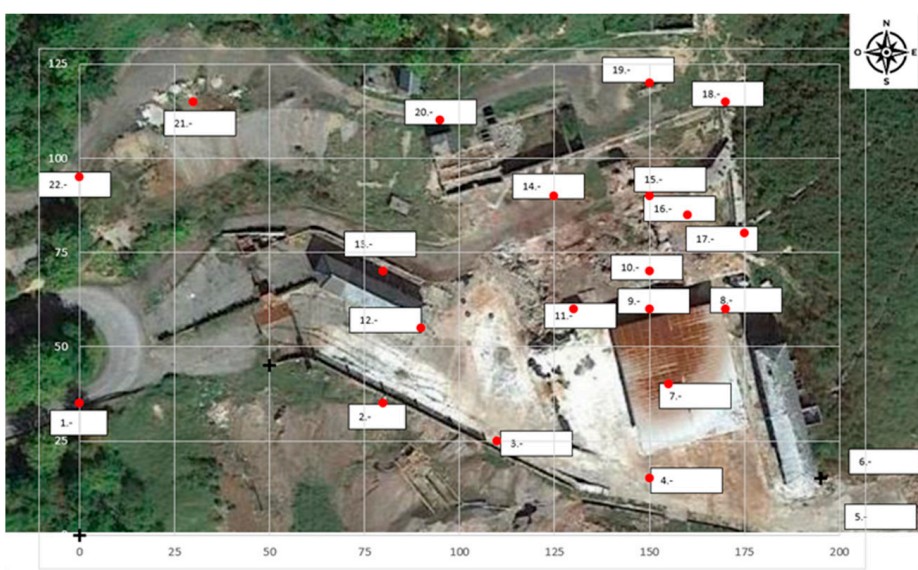

**Figure 4.** Site layout, (red points are locations or reading stations).

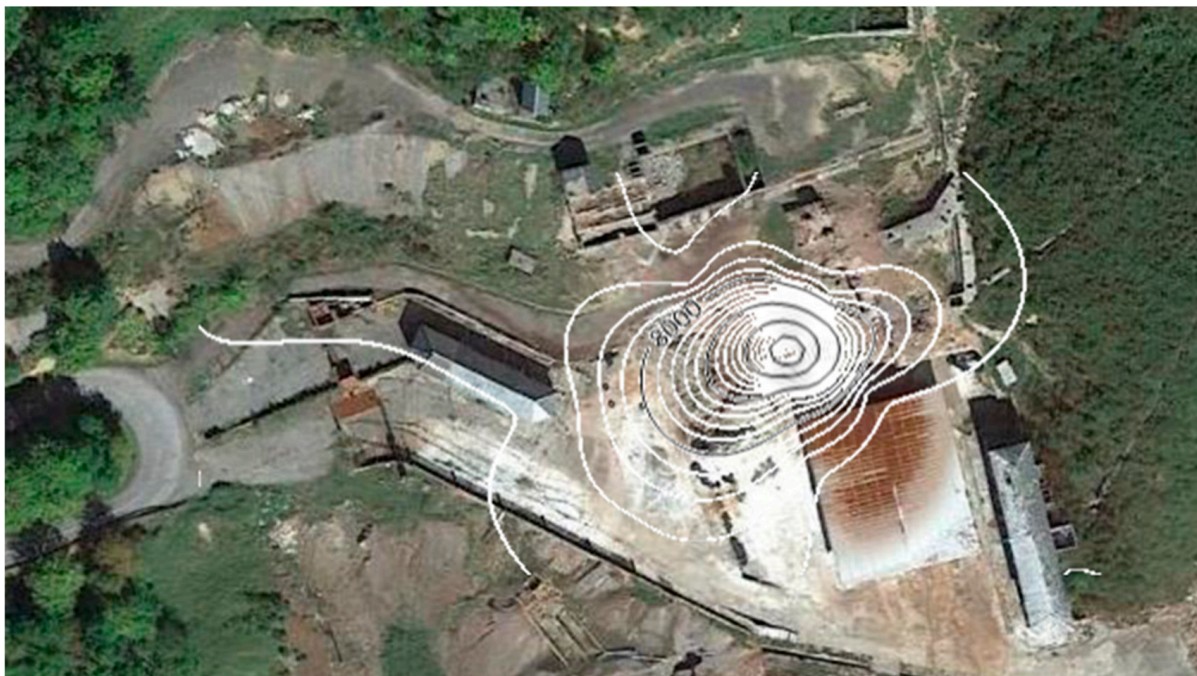

**Figure 5.** Survey 2's gaseous mercury concentration, (white lines are concentration isolines around the focus).

The maximum concentration was on the rubble of level 0, and it subsequently declined as we walked away from that location.

It appears evident that this area at level 0, where the debris from the demolition of the metallurgical plant buildings is located, was the source of gaseous mercury emissions that diffused or dispersed throughout the site.

### 3.1.4. Analysis of the Results at Representative Points

The final goal was to investigate the gaseous mercury contamination for occupational risk prevention in various portions of the site.

As a result, in addition to the global study, it was essential to conduct a detailed analysis of the contamination at various points to characterize the work at these locations in terms of risk, and four intervals were established to describe the level of contamination:

- Points where the concentration did not exceed 2000 ng/m$^3$ (10% of the OELV).
- Points where the concentration did not exceed 5000 ng/m$^3$ (25% of the OELV).
- Points where the concentration did not exceed 10,000 ng/m$^3$ (50% of the OELV).
- Points where the concentration could reach, and even exceed, 20,000 ng/m$^3$ (100% of the OELV).

Analysis of Points 2 and 3

Figure 6 depicts the concentrations of gaseous mercury at point 2 (Figure 6A) and point 3 (Figure 6B) as a function of the temperature.

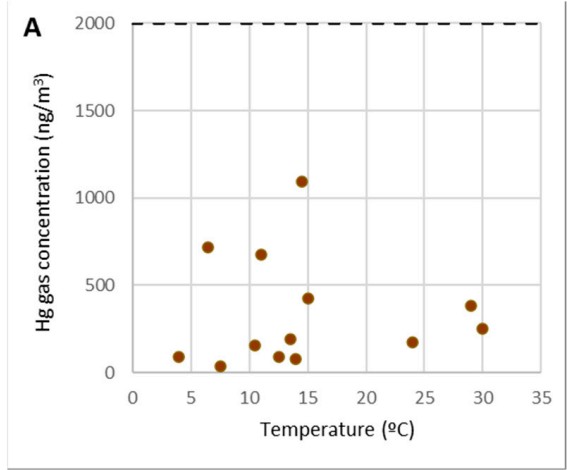 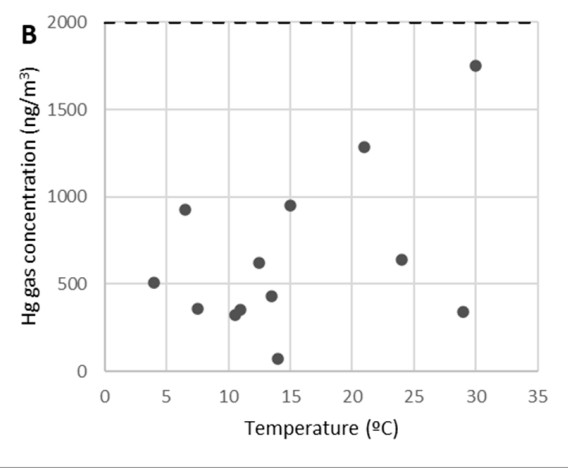

**Figure 6.** Concentrations of gaseous mercury at point 2 (**A**) and point 3 (**B**).

Points 2 and 3 (along with others such as 5 and 12) are examples of areas for visitors, locations for workers to rest, and passing zones.

As can be seen from the graphs, these are locations where gaseous mercury concentrations were consistently below 2000 ng/m$^3$ (10% of the OELV) and frequently below 500 ng/m$^3$. Within the SUBproducts4LIFE work area, in the areas with the lowest concentration (only at distant points such as 1 or 6), lower concentrations were found. As a result, in points 2 and 3 (as well as points 5 and 12), any work or activity, including the welcoming of visitors from outside the project, can be carried out at any time of the year.

It should be observed that there was no direct link between gas concentration and temperature, and it could be assumed that there is no gaseous mercury emission in them and that the gaseous mercury concentration in the air was caused by dispersion or diffusion from those points where there were emissions.

Analysis of Points 20 and 21

The work zones at level 2, including the former metallurgical plant waste dump, are represented by points 20 and 21 in Figure 7.

Despite the high mercury content found in the dump, these points did not emit considerable gaseous mercury emissions; hence, the concentration of gaseous mercury in these locations is on par with points 2 and 3. As a result, these are regions where any form of work or activity can be carried out without concerns about the amount of gaseous mercury in the environment. Furthermore, as previously stated, work may be performed at any time of year because the gas concentrations remain below 2000 ng/m$^3$, regardless of temperature.

The most significant concentrations (more than 1000 ng/m$^3$) were predominantly associated with wind gusts that disseminated the gaseous mercury from the demolition debris at level 0.

It is crucial to note that, in terms of contamination by gaseous mercury, work in the top dump (pilot case 1 in the SUBproducts4LIFE project) could be performed without significant restrictions.

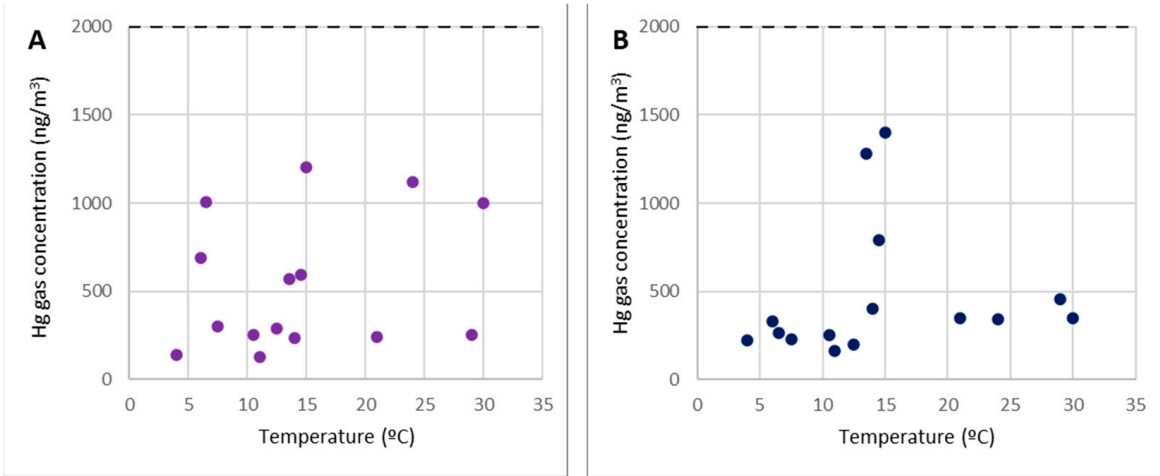

**Figure 7.** Concentrations of gaseous mercury at point 20 (**A**) and point 21 (**B**).

Analysis Points 4 and 7

Point 4 was where extra work was carried out, such as machinery maintenance or truck reception that transported the industry's by-products.

It was nearly identical to the previously examined locations, as shown in Figure 8A (especially points 2 and 3). Any activity could be carried out in this region without time constraints related to gaseous mercury. Gaseous mercury concentrations were always below 2000 ng/m$^3$ (10% of the OELV) and usually below 1000 ng/m$^3$.

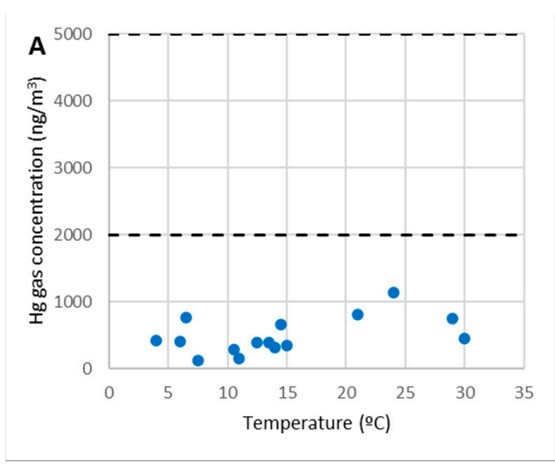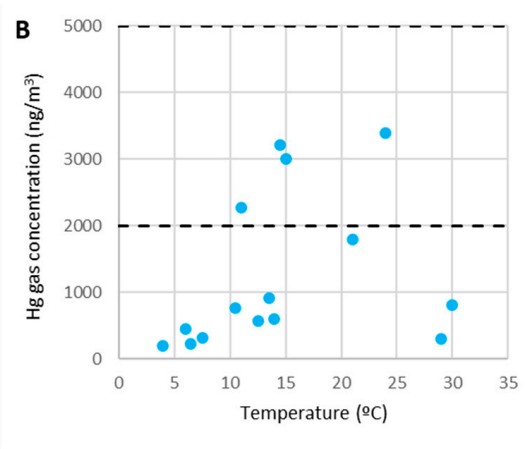

**Figure 8.** Concentrations of gaseous mercury at point 4 (**A**) and point 7 (**B**).

In the SUBproducts4LIFE project, point 7 was under the shelter of the filter channels, where pilot case 3 was being constructed. The concentration there could exceed the 2000 ng/m$^3$ limit for short periods, but the maximum concentration measured was always under 4000 ng/m$^3$. This point was closer to the demolition debris and more impacted by its emissions.

Because of its proximity to the contaminated rubble and the fact that it exceeded 10% of the OELV, it must be considered a higher-risk area. As a result, while would be feasible to stay in it, it should only be open to workers and not to the general public, and actions such as shortening the working time should be taken.

On the other hand, specific procedures such as using personal protective equipment (PPE) were already in place for carrying out the work near point 7.

Analysis of Points 8 and 11

Points 8 and 11 were approximately 20 m from the debris center and, along with point 9, formed a corridor leading to the debris region. Because of their closeness, gaseous mercury concentrations at both places were substantially higher than at the preceding points, even though they were below the OELV. However, owing to their location, they had to be in transit and were work areas associated with the SUBproducts4LIFE project's pilot case 2. As a result, access was strictly limited, and only project personnel with the necessary PPE and health and safety measures were allowed to enter.

As in the preceding situations, the temperature had no effect at points 8 and 11, and high concentrations could be found at very low temperatures, as could low concentrations at relatively high temperatures. A light breeze or a slight change in its direction caused significant variations in the concentrations at these locations; the concentration lowered or rose depending on whether the breeze blew towards or away from the contaminated material. As a result, the presence of gaseous mercury was more likely attributable to the migration (dispersion or diffusion) of gaseous mercury from the debris to that location.

The concentration was below 5000 ng/m$^3$ at point 8, which was protected from the debris by a high wall that functioned as a gas barrier (Figure 9A). However, at point 11, at the same distance but without an obstacle, it routinely exceeded that threshold, with concentrations of 5000 to 10,000 ng/m$^3$ (Figure 9B).

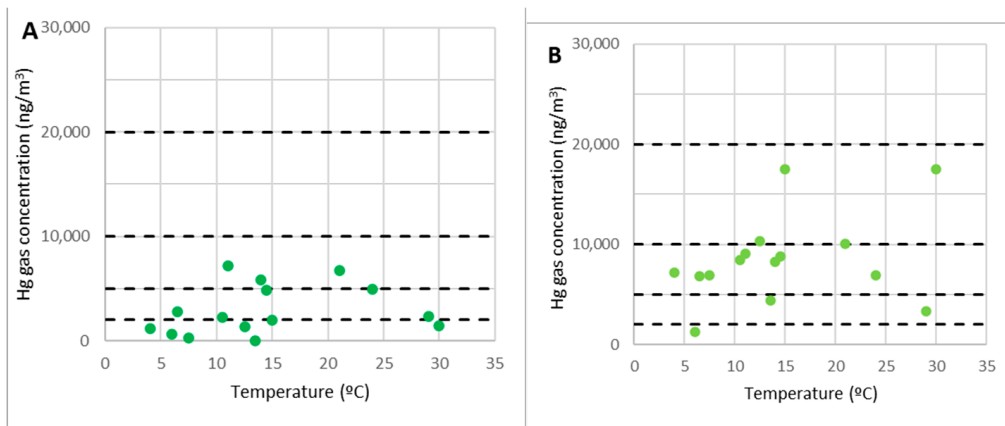

**Figure 9.** The concentrations of gaseous mercury in points 8 (**A**) and 11 (**B**).

One more consideration must be made to understand the results correctly. Part of the debris moved toward point 11 when the measurements were taken, reducing its distance from the rubble. As a result, when the concentrations should have been lower at low temperatures, the mercury gas concentration did not drop considerably since the debris was closer at that time.

There seem to be inconsistencies in other research conducted in 2007 in these adjacent locations. Loredo et al. reported maximum gaseous mercury concentrations of approximately 3000–3500 ng/m$^3$ [10]. The reason is that measuring stations in that campaign were further away from the rubble area. On the other hand, it must be taken into account that when the mercury was measured in 2007, it had been evaporated from the surface for years and the emissions had diminished. The debris cleanup, which occurred before the initial campaign described here, might have exposed more mercury to the air, boosting emissions.

Analysis of Points 16 and 17

Points 16 and 17 are on level 1, i.e., the intermediate rubble near the original chimney.

Point 16 is on demolition debris, but, although the concentration at that point fluctuated with temperature, there was no clear correlation between the two variables and it was

not clear that it was a significant emitting source. In terms of emissions, it was comparable to points 8 and 11 because the concentrations of gaseous mercury could occasionally reach the 10,000 ng/m$^3$ barrier, though they were usually below it (Figure 10A).

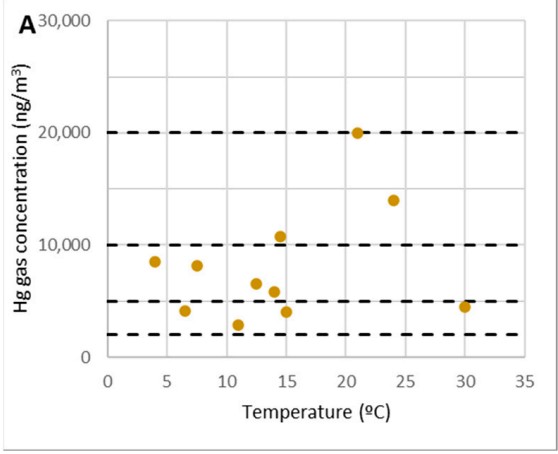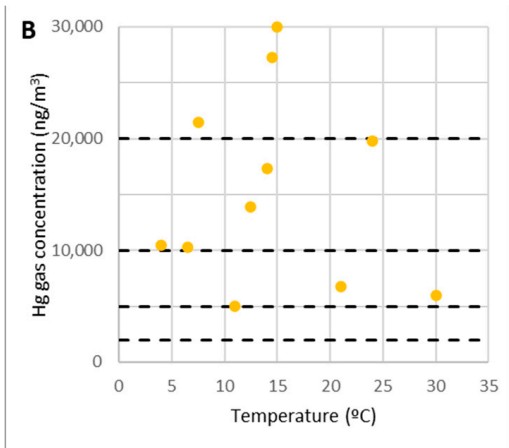

**Figure 10.** The concentration of gaseous mercury at points 16 (**A**) and 17 (**B**).

Point 17 was remarkable because it marked the start of the duct that directed smoke toward the chimney. It has been established that the confinement of air inside such conduits facilitates the attainment of high gaseous mercury concentrations. However, these concentrations were not reached outside of it, and so there was no clear relationship between temperature and concentration (Figure 10B). It was comparable to point 11. However, unlike the rubble of level 0, it did not appear to be an emitting focus. Because no work would be developed on these points, data were not collected from all surveys.

Analysis of Points 9 and 10

Point 10 was barely over level 0, on highly contaminated material, whereas point 9 was on the edge the accumulating rubble.

There was a clear relationship between the concentration of gaseous mercury in the air and the temperature at point 10. The concentration was less than 10,000 ng/m$^3$ (50 percent of the OELV) at low temperatures (less than 7 °C), when emissions were more negligible, while it exceeded 60,000 ng/m$^3$ at high temperatures (30 °C or higher) (OELV multiplied by three), as shown in Figure 11B.

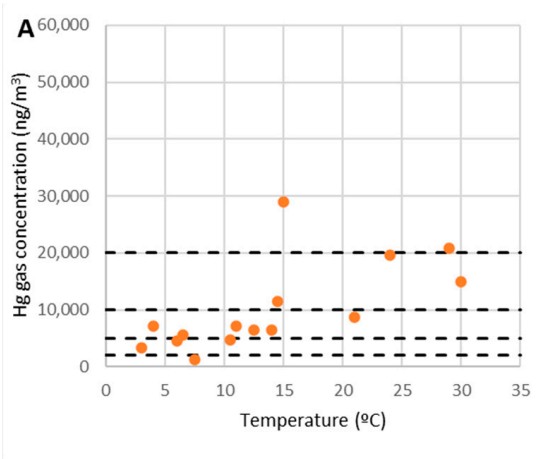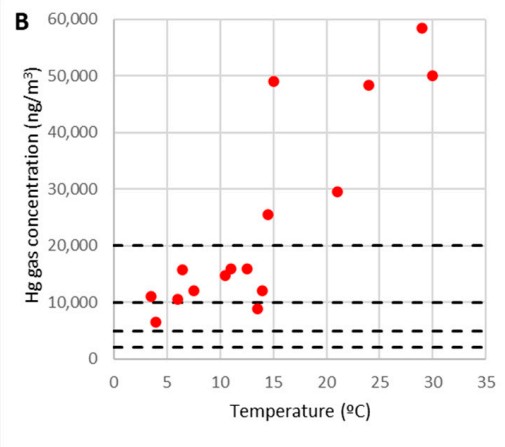

**Figure 11.** Concentrations of gaseous mercury at point 9 (**A**) and point 10 (**B**).

In general, gaseous mercury emissions rise with temperature; the link between the concentration at point 10 and temperature implied that point 10 at level 0 acted as a gaseous

mercury emitting source. Given these facts, it was evident that this was the most critical area. As such, when the ambient temperature exceeds 15 °C, work on it should not be completed continuously because the concentration might surpass the OELV of 20,000 ng/m$^3$.

Point 9 is on the edges of the rubble but not within it. It is similar to point 10 in that its concentration and temperature had a definite connection. However, there were discrepancies, such as the concentration not reaching the levels of the other location. At this stage, the concentration of gaseous mercury was always less than the OELV of 20,000 ng/m$^3$, and at average temperatures of 15 °C, it was 10,000 ng/m$^3$ or 50 percent of the OELV (Figure 11A). In other words, the concentration levels were similar to points 8 and 11, which were on the same corridor as the debris and the shelter, and the same recommendations would apply.

One interpretation for the difference in concentration at point 9 is that the presence of the gaseous mercury was due to pure diffusion from the rubble region outwards since it was precisely on the rubble's edge. Other elements that affect its focus include fluctuating breeze gusts, obstructions, and physical barriers in the remaining locations.

### 3.2. Application of Standard EN-689:2018

EN 689:2018 outlines the recommendations for a rigorous sampling procedure [42]. A SEG (similar exposure group) is a group of workers exposed to a chemical agent at an equivalent level while performing their tasks. The standard aims to determine if work in a comparable exposure group SEG is compatible with the OELV developed for work involving exposure to a chemical agent. The standard uses the occupational exposure limit value (OELV) instead of the OELV, although they are equivalent.

In this case, the SEG was the technician responsible for measuring the mercury gas concentration in the environment throughout the site. The chemical agent was the gaseous mercury, with an established OELV of 20,000 ng/m$^3$. The level of exposure would be defined by the average concentration of gaseous mercury in the environment for 8 h.

To determine if the work is compatible or conforms to that OELV, the standard establishes that measurements of exposure to the chemical agent need to be carried out. The standard allows three exposure measurements to be carried out when the exposure is between 10% and 20% of the OELV. However, when the exposure is expected to exceed 20% of the OELV, it establishes that a minimum of six measures and a statistical analysis must be made. In this case, an analysis must be carried out according to the second of the hypotheses.

The standard establishes that, at the least, a measurement must measure exposure for a minimum of 2 h to represent a full 8 h day.

In this case, the sample consisted of the weighted average concentration of gaseous mercury obtained from monitoring the gaseous mercury in the environment throughout the 22 points because this was the exposure level of the technician in charge of carrying out the task. Therefore, the duration of the sample collection was equal to the duration of the tour, which was approximately 1 h.

Although this period is less than the necessary two hours, there are several reasons why samples should not be made for longer than two hours:

It was necessary to monitor 22 points; thus, installing the equipment at each point for two hours was not an option.

To conduct an emissions study, monitoring the 22 control sites should take as little time as feasible so that the concentrations in all of the points are acquired under similar conditions. Weather conditions (particularly temperature) could fluctuate significantly over two hours.

As a first step, if there is no knowledge about the gaseous mercury at the site, it appears reasonable to shorten the route's duration by as much as possible. Following this, it was established that the measurements were representative despite our scenario's sampling duration being fewer than two hours.

A total of 15 control surveys are theoretically available for the statistical test. Assuming that there are n samples, being one sample, the weighted average concentration of gaseous

mercury $C_k$ was obtained from monitoring the gaseous mercury throughout the 22 points. In this case, n = 15 and k varies from 1 to 15. The 15 samples are ordered from lowest to highest. For each sample, the probability $p_k$ that the concentration is less than that of the sample $C_k$ is calculated as follows:

$$p_k = \frac{k - \frac{3}{8}}{n + \frac{1}{4}} \tag{1}$$

The measured values of the exposure $C_k$ are arranged in ascending order and plotted on the horizontal axis against the corresponding probabilities $p_k$ on the vertical axis on a log-probability paper. The good fit to a straight line shows that these results are distributed log-normally [42].

Representing the 15 points, Figure 12A verifies that 12 points effectively follow a straight line while three are separated. The points that do not follow the line correspond to two different situations:

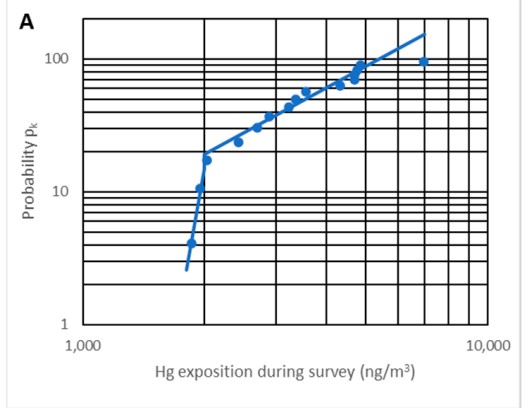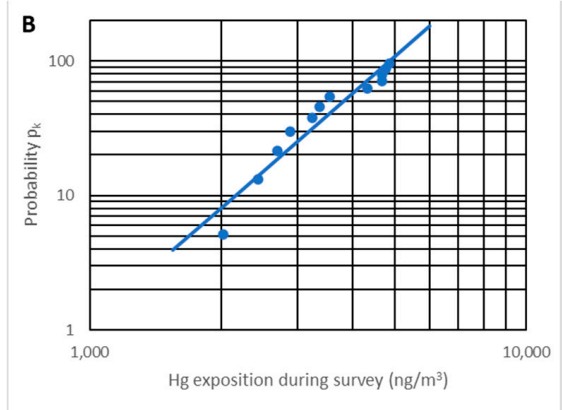

**Figure 12.** Representation of $p_k$ vs. $C_k$ (ng/m$^3$) for n = 15 (**A**) and n = 12 (**B**).

The maximum concentration point corresponds to the anomalous result found previously; it is not a representative result since no more was produced despite having made many more measurements, and because it is a non-representative point, it is not considered.

On the opposite side, two points correspond to the minimum value of exposure during a survey (approximately 2000 ng/m$^3$). Increasing the number of measurements, a set of points will appear on the vertical line $C_k$ = 2000 ng/m$^3$, and not being on the line that marks the trend, these two points are not taken into account. We note that these minimum values are approximately constant and approximately 10% of the OELV, and no other analysis is completed with them.

The data set utilized in the analysis is shown in Table 5, with the calculation of other variables described below.

Figure 12B depicts the pk probability versus the exposure $C_k$ (or average concentration for the entire route) on a log probability paper. It is proven that there is a very high linear correlation ($r^2$ = 0.92), implying that the distribution is truly log-normal.

The geometric mean GM and geometric standard deviation GSD must be determined from these data using the following formulas:

$$\ln(GM) = \frac{\sum_1^n \ln(C_k)}{n} \quad GM = \exp\left(\frac{\sum_1^n \ln(C_k)}{n}\right) \tag{2}$$

$$\ln(GSD) = \sqrt{\frac{\sum_1^n (\ln(C_k) - \ln(MG))^2}{n-1}} \quad GSD = \exp\left(\sqrt{\frac{\sum_1^n (\ln(C_k) - \ln(MG))^2}{n-1}}\right) \tag{3}$$

**Table 5.** Calculation for n = 12.

| K. | $C_k$ (ng/m³) | $p_k$ | $p_k$ (%) | $\ln(C_k)$ | $[\ln(C_k) - \ln(MG)]^2$ |
|---|---|---|---|---|---|
| 1 | 2025 | 0.05 | 5.10 | 7.61 | 0.29 |
| 2 | 2426 | 0.13 | 13.27 | 7.79 | 0.13 |
| 3 | 2692 | 0.21 | 21.43 | 7.90 | 0.07 |
| 4 | 2882 | 0.30 | 29.59 | 7.97 | 0.04 |
| 5 | 3230 | 0.38 | 37.76 | 8.08 | 0.01 |
| 6 | 3359 | 0.46 | 45.92 | 8.12 | 0.00 |
| 7 | 3552 | 0.54 | 54.08 | 8.18 | 0.00 |
| 8 | 4317 | 0.62 | 62.24 | 8.37 | 0.05 |
| 9 | 4679 | 0.70 | 70.41 | 8.45 | 0.09 |
| 10 | 4682 | 0.79 | 78.57 | 8.45 | 0.09 |
| 11 | 4755 | 0.87 | 86.73 | 8.47 | 0.10 |
| 12 | 4859 | 0.95 | 94.90 | 8.49 | 0.11 |

This test is based on comparing the 70% upper confidence limit (UCL) with the 95th percentile of the distribution of the results. The UCL is calculated using the geometric mean (GM) and geometric standard deviation (GSD).

From these data, the variable $U_R$ is calculated:

$$U_R = \frac{\ln(OELV) - \ln(GM)}{\ln(GSD)} \tag{4}$$

This value must be verified with the $U_T$ variable and tabulated according to n (Table 6). If $U_R \geq U_T$, then the conclusion is compliance with the OELV.

If $U_R < U_T$, then the conclusion is non-compliant with the OELV.

From the data of the 12 control surveys, it was found that $U_R = 5.90$ is greater than $U_T = 1.961$ (n = 12), and therefore, there is compliance with the OELV = 20,000 ng/m³.

**Table 6.** UT variable tabulation according to UNE-EN-689:2019 standard.

| n | $U_T$ | n | $U_T$ | n | $U_T$ |
|---|---|---|---|---|---|
| 6 | 2.187 | 15 | 1.917 | 24 | 1.846 |
| 7 | 2.12 | 16 | 1.905 | 25 | 1.841 |
| 8 | 2.072 | 17 | 1.895 | 26 | 1.836 |
| 9 | 2.035 | 18 | 1.886 | 27 | 1.832 |
| 10 | 2.005 | 19 | 1.878 | 28 | 1.828 |
| 11 | 1.981 | 20 | 1.87 | 29 | 1.824 |
| 12 | 1.961 | 21 | 1.863 | 30 | 1.82 |
| 13 | 1.944 | 22 | 1.857 | | |
| 14 | 1.929 | 23 | 1.851 | | |

This result supports that the gaseous mercury concentration monitoring work could be performed even for eight consecutive hours. In other words, these results would also justify the possibility of developing other tasks on the site, for example, taking samples of gaseous mercury concentrations in the environment, taking water samples, visiting the site to plan work, carrying out preparatory work, etcetera. Work in the rubble would require a more specific investigation that is not the objective of this paper.

### 3.3. Development of an Empirical Model

The study's primary goal was to anticipate the gaseous mercury concentrations in particular locations to organize activities in those areas to avoid occupational hazards. The target was to predict the concentrations in those points. As a result, an empirical model based on the field data is provided, leaving the development of an analytical model based on chemical and physical principles of gas diffusion and pollutant dispersion in the atmosphere to another study.

3.3.1. Concentration of Gaseous Mercury in the Highly Contaminated Rubble

From the initial research, it seems that there is a considerable concentration of metal mercury in the area of the highly contaminated demolition debris from the metallurgical plant (point 10) which is capable of evaporating and being emitted into the atmosphere, acting as an emitting source of gaseous mercury. As a result, a more detailed investigation of points 9 and 10 is feasible. While the temperature was 15 °C, a concentration of 30,000 ng/m$^3$ was recorded on the rubble in campaign number 3. This result was not replicated because it was deemed abnormal, and it was excluded from the analysis.

The data could be used to develop an analytical model based on the physical chemistry of the phenomenon, but that is an independent line of investigation. Assuming the initial state of the project, an empirical model is suitable for occupational risk analysis because it is easy to obtain from measurements, easy to use, and allows estimates to be made quickly.

A regression line $\ln(C_{10}) = c_1 + c\,\theta$ is fitted to the experimental data corresponding to point 10 to determine the dependency of the gas concentration on the debris $C_{10}$ (ng/m$^3$) with the temperature (°C), yielding the following exponential relationship with a correlation coefficient of 0.81 (Figure 13B):

$$C_{max} = C_{10} = 6759\,e^{0.0704\,\theta} \tag{5}$$

This equation is temperature-dependent and only applies to the region above the debris, i.e., r < R, where R is the distance between the rubble area's center and the edge.

A similar link can be seen at point 9, which was on the outside of the debris area (Figure 13A):

$$C_9 = 2917\,e^{0.0635\,\theta} \tag{6}$$

According to the measurements, the concentration at point 9 was approximately 36% of the highest concentration at point 10. It can be assumed as a rough approximation:

$$C_9 \approx \frac{C_{max}}{2.65} \tag{7}$$

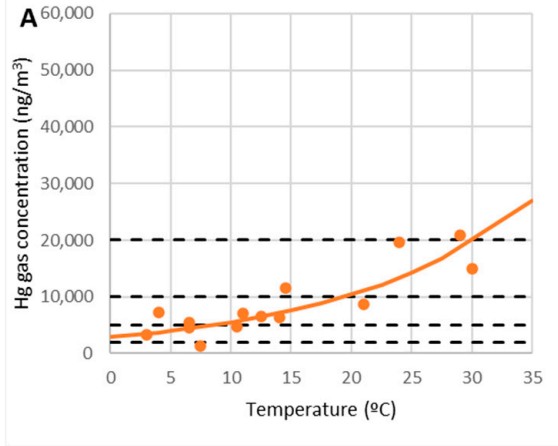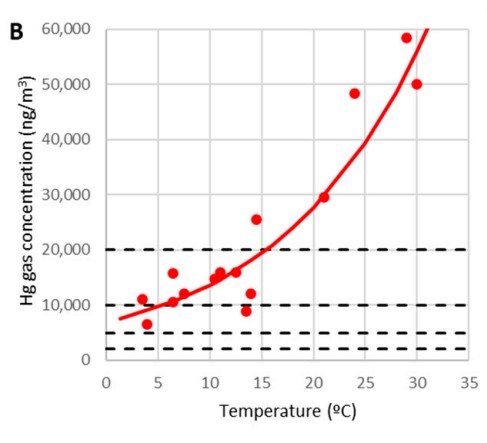

**Figure 13.** Model of gaseous mercury concentrations at points 9 (**A**) and 10 (**B**).

### 3.3.2. Distribution of the Concentrations of Gaseous Mercury in the Work Points Surrounding the Focus

At the limit of the focus, point 9, concentrations of up to 20,000 ng/m$^3$ were measured, similar to those found by Loredo et al. [10] and Cabassi et al. [13] in other similar facilities. However, they are much larger than those measured by Qiu et al. [12], which were approximately 400 ng/m$^3$, possibly because they were not mineral processing facilities. The present research indicates that the rubble area has a much higher emission potential than mineral waste disposal does.

The mercury gas concentration at the other points on the site varies with the distance to the center of the demolition rubble, indicating that there were no considerable gaseous mercury emissions, but rather that they arrived via diffusion or dispersion in the atmosphere.

Because the goal was to forecast the concentrations at the points where work may be performed, points 13, 14, 15, 16, and 17 are not be addressed in the following. Point 11 is also not considered due to its distance variation from the rubble.

When the concentration at each location C (ng/m$^3$) is plotted against the distance to the rubble's center r (m), it can be seen that the concentration drops as the length decreases:

$$C(r) \approx \frac{K}{r} \tag{8}$$

The graphs in Figure 14 show the adjustments of the point clouds to this law for the temperature intervals $\theta$ = 0–15 °C (A) and $\theta$ = 15–30 °C (B). The correlation coefficients obtained are 0.83 and 0.79, respectively, and so it can be concluded that this hypothesis is acceptable.

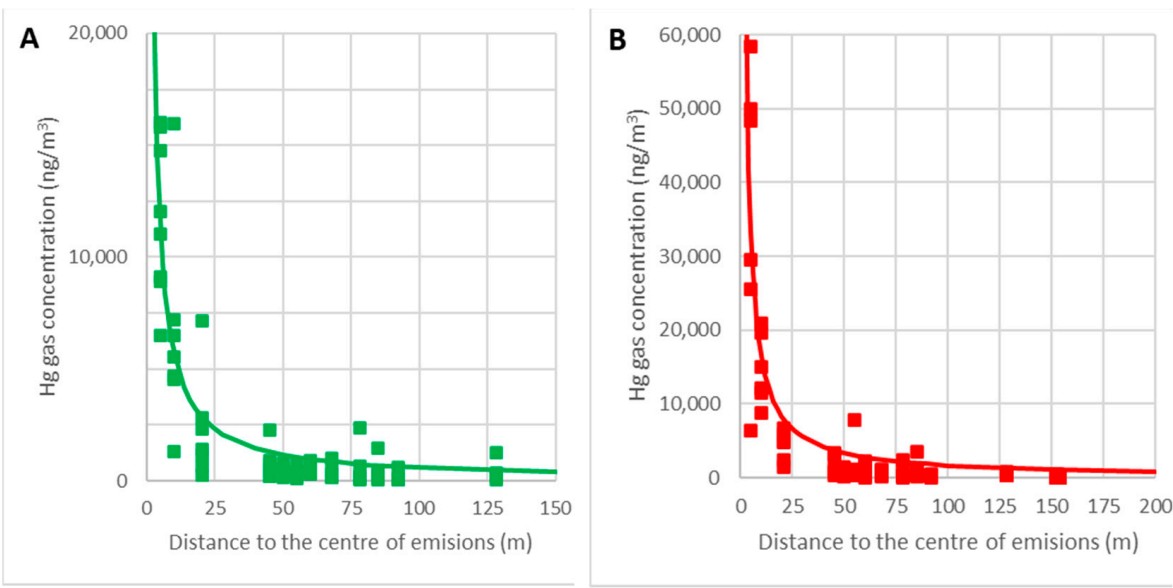

**Figure 14.** Variations in the gaseous mercury concentrations with distance for temperatures of 0–15 °C (**A**) and 15–30 °C (**B**).

Given that the concentration near the edge (point 9) was approximately $C_{max}$/2.65, at every point r > $r_0$ away from the center of the debris, we can obtain:

$$C(r) \approx \frac{C_{max}}{2.65} \left( \frac{r_0}{r} \right) \tag{9}$$

As stated, $r_0$ is the distance from the center to the edge of the rubble area. Since the distance from point 9 to the center of the rubble was 10 m, $r_0$ = 10 m is taken, and so the

empirical model of the concentrations of gaseous mercury at the work points of the site (r >10 m) would be:

$$C(r) = 2550 \, e^{0.0704 \, \theta} \left( \frac{10}{r} \right) \tag{10}$$

The graphs in Figure 15 represent the decrease in the concentration of gaseous mercury as a function of the distance to the center of the debris. Real data for the ambient temperatures $\theta$ = 5–10 °C (A) and $\theta$ = 10–15 °C (B) are represented, with the lines corresponding to the calculated values for $\theta$ = 10 °C (A) and $\theta$ = 15 °C (B).

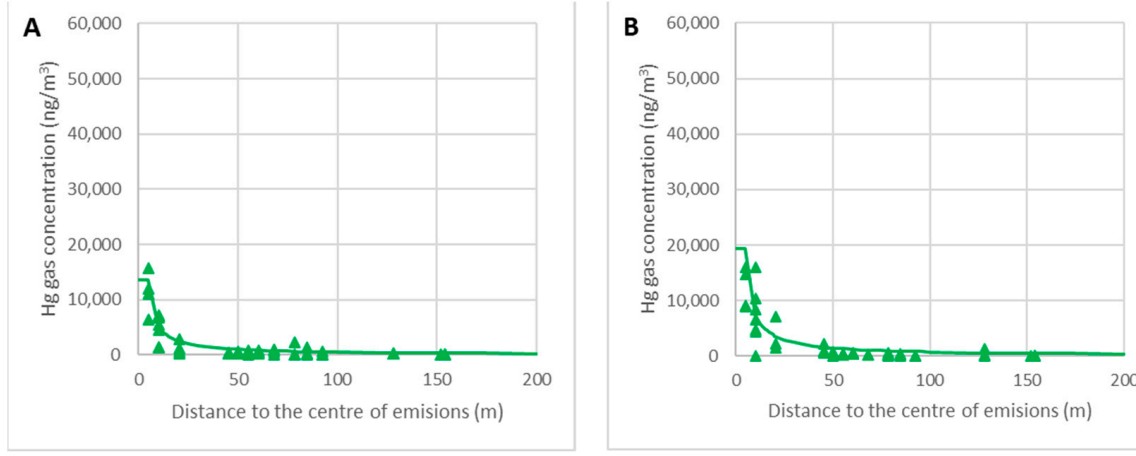

**Figure 15.** Variations in the gas concentrations around the focus for $\theta$ = 5–10 °C (**A**) and $\theta$ = 10–15 °C (**B**).

In Figure 16, real data for the ambient temperatures $\theta$ = 15–20 °C (A) and $\theta$ = 25–30 °C (B) are represented, with the lines corresponding to the calculated values for $\theta$ = 20 °C (A) and $\theta$ = 30 °C (B).

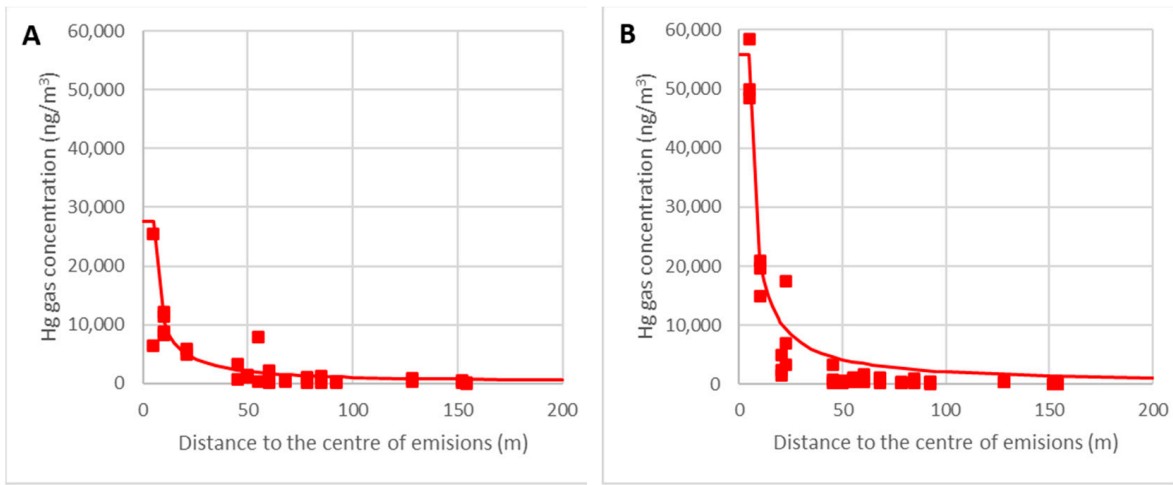

**Figure 16.** Variations in the gas concentrations around the focus for $\theta$ = 15–20 °C (**A**) and $\theta$ = 25–30 °C (**B**).

It can be verified that the model predicts the concentration as a function of the ambient temperature with enough accuracy, which is critical for work schedules. The most crucial area was located within the nearest 50 m because the limit of 5000 ng/m$^3$ could be reached with high temperatures. When further away than 50 m, the empirical model is less accurate, but this is not relevant because the gaseous mercury concentration is very low and the model overestimates the mercury gas concentrations as being on the safe side.

While the gaseous mercury concentration value may be anticipated, monitoring the actual concentration when working in the highest-risk region is vital for ensuring the health and safety of workers.

### 3.3.3. Temperature-Based Representation of the Distinct Zones' Extension

This empirical model provides a global understanding of how emissions are generated and how the distribution of gaseous mercury concentrations at a site can be obtained. It has been established that temperature is the essential variable to consider in the case of atmospheric stability.

Applying the model to the site in Figure 17, it is easy to locate different zones with different risks for exposure to gaseous mercury: the red line is an area with more than 100% of the OELV (20,000 ng/m$^3$), the orange line is an area with more than 50% of the OELV (10,000 ng/m$^3$), the green line is an area within which the concentration is greater than 25% of the OELV (5000 ng/m$^3$), and, finally, the blue line is an area within which the concentration is greater than 10% of the OELV (2000 ng/m$^3$).

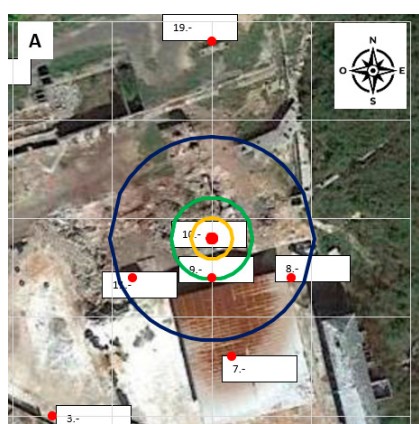 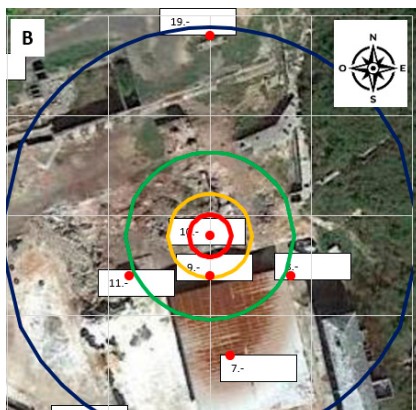 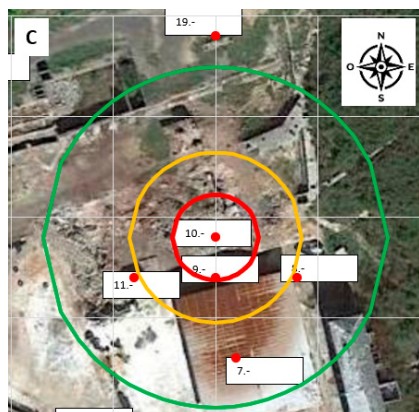

**Figure 17.** Zones determined with the empirical model for $\theta$ = 10 °C (**A**), $\theta$ = 20 °C (**B**), and $\theta$ = 30 °C (**C**). (Red circle C > 20,000 ng/m$^3$, orange circle C > 10,000 ng/m$^3$, green circle C > 5000 ng/m$^3$, blue circle C > 2000 ng/m$^3$.).

Figure 17 shows the zones determined with the model for $\theta$ = 10 °C, $\theta$ = 20 °C, and $\theta$ = 30 °C. The concentration of 5000 ng/m$^3$ (25% of the OEVL) can be taken as a reference for safe working conditions because there is compliance with the OELV= 20,000 ng/m$^3$.

The conclusion is straightforward: according to the model, at temperatures below 10 °C, there is no place where the concentration exceeds the OELV, making these the safest circumstances for working in the rubble. There is a transition between 15 °C and 20 °C where concentrations higher than the OELV appear in the debris; outside the rubble, the conditions are suitable for routine work. Conditions deteriorate as temperatures rise, but only in the rubble area and the few meters surrounding it, where concentrations can be extremely high. Work compliant with the legal limit of OELV = 20,000 ng/m$^3$ could be carried out in most of the La Soterraña mining facility.

### 3.3.4. The Model's Application in Planning

The model's main benefit is that it allows for the assessment of various scenarios and can assist project planners in calculating the corresponding exposures and making preliminary work plans. It is simple to use the model to demonstrate that working on the rubble at temperatures of below 10 °C can be completed in an 8 h workday. Due to workers having to wear masks, rest is compulsory after working for two hours. The following planning tasks can be taken as an example (Table 7):

**Table 7.** Examples of planning tasks.

| Task | Location | Hours |
|------|----------|-------|
| 1 | Work within the rubble (point 10) | 2 |
| 2 | Break (point 3) | 1 |

**Table 7.** *Cont.*

| Task | Location | Hours |
|------|----------|-------|
| 3 | Work within the rubble (point 10) | 2 |
| 4 | Lunch time (no exposure) | 1 |
| 5 | Work within the rubble (point 10) | 2 |
| 6 | Machinery maintenance work | 1 |

The model can forecast the level of gaseous mercury exposure at any temperature at any point. We can assume that the model results represent the exposure levels in the first approach. To have a representative set of values, it is assumed that the temperature varies between 10 °C and 15 °C for six consecutive days. When working with temperatures between 10 °C and 15 °C (Table 8), an equivalent day exposure using a weighted average can be obtained. The overall result is 12,580 ng/m$^3$, 63% of the OELV = 20,000 ng/m$^3$.

**Table 8.** Gaseous mercury concentrations from 10 °C to 15 °C in the example location.

| $\theta$ (°C) | $C_{10}$ (ng/m$^3$) | $C_3$ (ng/m$^3$) | $C_{10}$ (ng/m$^3$) | $C_{10}$ (ng/m$^3$) | $C_4$ (ng/m$^3$) | $C_{eq}$ (ng/m$^3$) |
|------|------|------|------|------|------|------|
| 10 °C | 13,666 | 859 | 13,666 | 13,666 | 937 | 10,474 |
| 11 °C | 14,662 | 922 | 14,662 | 14,662 | 1006 | 11,238 |
| 12 °C | 15,732 | 989 | 15,732 | 15,732 | 1079 | 12,057 |
| 13 °C | 16,879 | 1061 | 16,879 | 16,879 | 1158 | 12,937 |
| 14 °C | 18,110 | 1139 | 18,110 | 18,110 | 1242 | 13,880 |
| 15 °C | 19,431 | 1222 | 19,431 | 19,431 | 1333 | 14,893 |

Applying the statistical test of the EN 689 standard to the six indicated cases, it is possible to verify that there is a log-normal distribution with r$^2$ = 0.91 (Figure 18A). When applying the statistical analysis, the result is UR = 6.24, which is more significant than UT = 2.187 (n = 6), and therefore there is compliance with the OELV = 20,000 ng/m$^3$. As the mercury emissions are lower for lower temperatures, working with temperatures of less than 10 °C also complies with the OELV.

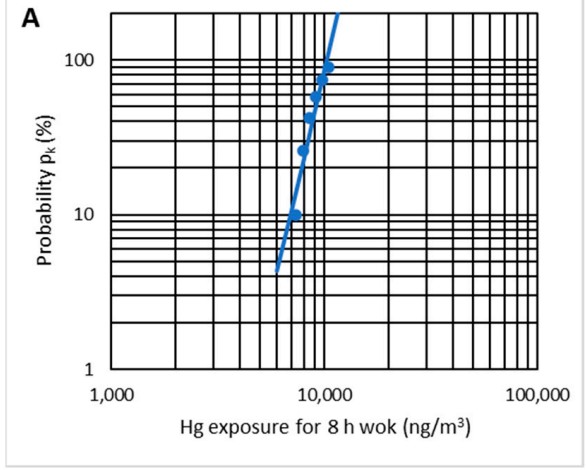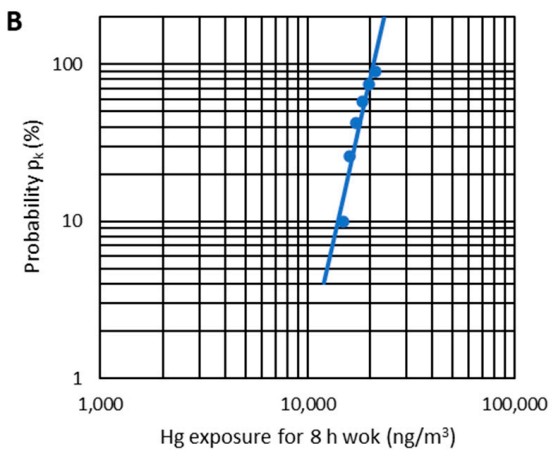

**Figure 18.** Log-normal distribution check-up for temperatures of 5–10 °C (**A**) and 15–20 °C (**B**).

If we repeat the process with temperatures of above 15 °C, it would be seen that it is no longer possible to work 8 h following the same schedule and that the working time within the rubble would have to be reduced during the day. The model allows for estimating the gaseous mercury concentrations at the operational points on six different days with

temperatures varying from 15 °C to 20 °C (Table 9). On average, the overall equivalent day exposure is 17,887 ng/m$^3$, 89% of the OELV.

**Table 9.** Gaseous mercury concentrations from 15 °C to 20 °C in the example location (alternative 1).

| θ (°C) | $C_{10}$ (ng/m$^3$) | $C_3$ (ng/m$^3$) | $C_{10}$ (ng/m$^3$) | $C_{10}$ (ng/m$^3$) | $C_4$ (ng/m$^3$) | $C_{eq}$ (ng/m$^3$) |
|---|---|---|---|---|---|---|
| 15 °C | 19,431 | 1222 | 19,431 | 19,431 | 1333 | 14,893 |
| 16 °C | 20,848 | 1311 | 20,848 | 20,848 | 1430 | 15,979 |
| 17 °C | 22,369 | 1407 | 22,369 | 22,369 | 1534 | 17,144 |
| 18 °C | 24,001 | 1509 | 24,001 | 24,001 | 1646 | 18,395 |
| 19 °C | 25,751 | 1619 | 25,751 | 25,751 | 1766 | 19,737 |
| 20 °C | 27,629 | 1737 | 27,629 | 27,629 | 1895 | 21,176 |

By applying the EN 689 standard's statistical test, we can verify a log-normal distribution with r$^2$ = 0.91 (Figure 18B). In this case, contrary to the previous one, the variable $U_R$ = 0.902 is lower than $U_T$ = 2.187 (n = 6); therefore, there is no compliance with the OELV = 20,000 ng/m$^3$.

The solution is to reduce the time spent working at the demolition debris area. For example, let us assume we are operating at points 21 and 4 during the afternoon. In this case, the exposure is reduced significantly (Table 10). On average, the overall equivalent day exposure is 12,225 ng/m$^3$, 61% of the OELV. After the statistical analysis, the parameter $U_R$ = 3.792 is greater than $U_T$ = 2.187 (n = 6), which means that, effectively, there is compliance with the OELV = 20,000 ng/m$^3$.

**Table 10.** Gaseous mercury concentrations from 15 °C to 20 °C in the example location (alternative 2).

| Time (h) | 2 | 1 | 2 | 2 | 1 | 8 |
|---|---|---|---|---|---|---|
| θ (°C) | $C_{10}$ (ng/m$^3$) | $C_3$ (ng/m$^3$) | $C_{10}$ (ng/m$^3$) | $C_{21}$ (ng/m$^3$) | $C_4$ (ng/m$^3$) | $C_{eq}$ (ng/m$^3$) |
| 15 °C | 19,431 | 1222 | 19,431 | 573 | 1333 | 10,178 |
| 16 °C | 20,848 | 1311 | 20,848 | 614 | 1430 | 10,921 |
| 17 °C | 22,369 | 1407 | 22,369 | 659 | 1534 | 11,717 |
| 18 °C | 24,001 | 1509 | 24,001 | 707 | 1646 | 12,572 |
| 19 °C | 25,751 | 1619 | 25,751 | 759 | 1766 | 13,489 |
| 20 °C | 27,629 | 1737 | 27,629 | 814 | 1895 | 14,472 |

It must be pointed out that these results are significant from a scientific point of view, and the developed model is a handy tool for analyzing conditions and planning tasks with a high level of safety before starting work.

Nevertheless, once the model is developed, new measures must be completed for 2 h. Then, the model must be recalibrated and used more accurately and within the legal requirements of the relevant standards. Although it is not in the scope of this research, the model's validity and results have been tested with a set of measures for 2 h.

## 4. Conclusions

Mercury mine facilities can be an occupational hazard with respect to mercury gas emissions. The empirical model developed in this paper can predict the gaseous mercury exposure for the workers, and it is a valuable tool for planning work. Following a practical model developed, all the Soterraña mine restoration works have been scheduled with a safe level regarding the gaseous mercury occupational hazards, avoiding surpassing the OELV and minimizing worker exposure to gaseous mercury.

In the same way, as the human body retains approximately 80% of inhaled mercury, the empirical model is a valuable technique for reducing worker exposure to mercury by selecting days or hours with the minimum temperature for carrying out work, with high standards in occupational hygiene.

The research confirms that in the studied location, there are areas where the gaseous mercury concentration is affected by the temperature (points 9 and 10); these points are the more contaminated areas.

The empirical model confirms that below 15 °C, it is possible to work a whole shift in any area of the mining facility without time restrictions; for temperatures above 15 °C, time restrictions must be applied in the rubble area (points 9 and 10).

The empirical model established that there is no risk of exposure to gaseous mercury for pedestrians or for the populations of the nearby villages from the emissions of the mining facility.

After validation, this model can be used in other mine facilities in Spain or in other parts of the world for planning the restoration of mines, minimizing the worker exposure to mercury gas and avoiding health hazards.

**Author Contributions:** Conceptualization, R.R. and H.G.-G.; data curation, R.R., H.G.-G. and E.G.-O.; investigation, R.R. and H.G.-G.; methodology, R.R., H.G.-G. and E.G.-O.; resources, E.G.-O.; supervision, R.R.; validation, R.R., H.G.-G. and E.G.-O.; writing—original draft, R.R. and H.G.-G. All authors have read and agreed to the published version of the manuscript.

**Funding:** The authors would like to thank the program LIFE of the European Commission for the funding received for the project SUBproducts4LIFE (reference LIFE16 ENV/ES/000481).

**Institutional Review Board Statement:** Not applicable.

**Informed Consent Statement:** Not applicable.

**Data Availability Statement:** The article provides all the data used in this research.

**Acknowledgments:** The authors would like to thank the institutions and private companies that participated in the project SUBproducts4LIFE: Biosfera consultoría Medioambiental (BIOSFERA), Escorias y Derivados (EDERSA), Global Service (GService), Hidroeléctrica del Cantábrico (EDP), Instituto Asturiano de Prevención de Riesgos Laborales (IAPRL), Recuperación y Renovación (R&R), and Universidad de Oviedo (UNIOVI). Finally, the authors thank their sponsors Arcelor Mittal, Ingeniería de Montajes Norte SA (IMSA), Asturbelga de Minas, and Lena Council, and the Instituto Nacional de Silicosis (INS) is also greatly appreciated.

**Conflicts of Interest:** The authors declare no conflict of interest.

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
