# Peer review of "Empirical Model of Gaseous Mercury Emissions for the Analysis of Working Conditions in Outdoor Highly Contaminated Sites"

_sustainability, doi:10.3390/su142113951_

Round 1
Reviewer 1 Report
General comments:
1. The manuscript does not follow the standard writing format of the journal.
2. The manuscript is written like a project progress or terminal report. It needs improvement by simplifying and writing it in paragraph form.
3. Follow the journal article format (Introduction, Materials and Methods, Results and Discussion, Conclusion, Recommendation)
4. Description of the mine (section 1.1) should be included in the Materials and Methods section.
5. To simplify the presentation of the site, there should be a Table presenting the sampling points and its corresponding site/area description (e.g within the former ball mill; metallurgy building, etc.)
6. The objectives of the paper should be mentioned in the last part of Introduction.
7. Citing of references should be in reference number [no.].
8. Figure 5 can not be understood by readers, this map should be underlayed by the locations/structures in the project site.
9. Conclusion must be written in paragraph form.
10. Needs to be consistent in writing Hg-gas and gaseous mercury. I suggest to use gaseous Hg in the entire manuscript instead of Hg-gas.
Reviewer 2 Report
Dear Editors,
The submitted Paper entitled " Empirical model of Hg emissions for the Analysis of the Working Conditions in Outdoor High Contaminated Sites" fits the scope of the Sustainability journal and presents an interesting study, with structured ideas.
The manuscript deals with an important topic. The structure is clear and logical. Before, making my overall decision, I read the manuscript many times to be confident on the review. I believe that the manuscript in its current form cannot be directly accepted for publication because it requires Minor changes. I hope that the authors do not get disappointed by this decision. The main reasons of my review are given below.
1- In the whole paper is often hard to guess what the author/authors had in mind.
2- Introduction: provide some more references to support key statements (from recent literature). Also, describe clearly your study objectives towards the end and connect those better to the state of the art you covered before in the same section.
3- Please give the study area in a separate section
4- It should be explained in detail how the individual parameters were obtained in the chapter "material and methods".
5- Discussion: would be nice if you could include a bit more critical reflection on the results obtained, also linking your study findings to the literature, and attempting to provide a deeper interpretation of your results.
6- Conclusions: I would like to have read punchier conclusions and a much clear provision of the study’s take-home message.
7- Please add a north arrow and scale to the right part of figures 1, 4 and 17
8- Please number the references in order of appearance in the text and list them individually at the end of the manuscript (according to the journal style)
9- Please number the equations in order of appearance in the text
10- In lines (294, 437, 442, 455, and 467)...the references in the text must be changed (ex: line 294, fig. 4 to figure 4)
11- There are a lot of panels. Please use sub-indications a, b, c...
12- In lines (318, 319, 563, 564, and 565), please respect the font size and style
13- Please follow the journal style.
Round 2
Reviewer 1 Report
Very good revision of the manuscript.